# Primary health care and social isolation against COVID-19 in Northeastern Brazil: Ecological time-series study

**Sanderson José Costa de Assis**[1]ᴏ*, **Johnnatas Mikael Lopes**[2]ᴏ, **Marcello Barbosa Otoni Gonçalves Guedes**[3]ᴏ, **Geronimo José Bouzas Sanchis**[1]ᴏ, **Diego Neves Araujo**[4]ᴏ, **Angelo Giuseppe Roncalli**[1]ᴏ

**1** Public Health Program, Universidade Federal do Rio Grande do Norte, Natal, Rio Grande do Norte, Brazil, **2** Medicine Department, Universidade Federal do Vale do São Francisco, Bahia, Brazil, **3** Physical Therapy Department, Universidade Federal do Rio Grande do Norte, Natal, Rio Grande do Norte, Brazil, **4** Medicine Department, College of Health Sciences, Paraíba, Brazil

ᴏ These authors contributed equally to this work.
* sanderson_assis@hotmail.com

## Abstract

**Data Availability Statement:** All relevant data are within the manuscript and its Supporting information files.

### Background

Brazil is witnessing a massive increase of corona virus disease (COVID-19). Its peculiar primary health care (PHC) system faces a burden due to the contagion occurring in the community environment. Then, the aim is to estimate the effect of the coverage of primary health care and social isolation on the evolution of confirmed cases and deaths by COVID-19, controlling sociodemographic, economic and health system aspects.

### Methods

A time series design was designed with data on diagnosed cases of COVID-19 and their deaths as outcomes in the capital cities of the Northeast region of Brazil. Independent variables such as PHC coverage, hospital beds, social isolation, demographic density, Gini index and other indicators were analyzed. A Autoregressive Generalized Linear Model method was applied for model the relationship.

### Results

We identified an exponential growth of cases ($y = 0.0025^{0.71x}$; *p-value*<0,001). However, there is a high variability in the occurrence of outcomes. PHC coverage$\geq$75% ($\chi2 = 9.27$; *p-value* = 0.01) and social isolation rate ($\chi2 = 365.99$; *p-value*<0.001) proved to be mitigating factors for the spread of COVID-19 and its deaths. Capitals with hospital beds $\geq$ 3.2 per thousand inhabitants had fewer deaths ($\chi2 = 9.02$; *p-value* = 0.003), but this was influenced by PHC coverage ($\chi2 = 30,87$; *p-value*<0.001).

**Funding:** The authors received no specific funding for this work.

**Competing interests:** The authors have declared that no competing interests exist.

## Conclusions

PHC mitigates the occurrence of Covid-19 and its deaths in a region of social vulnerability in Brazil together with social isolation. However, it is not known until when the system will withstand the overload in view of the low adhesion to social isolation, the lack of support and appropriate direction from the government to its population.

## Introduction

The new coronavirus disease (COVID-19) outbreak in the world culminated in the Public Health Emergency Declaration of International Importance by the World Health Organization [1], on January 30th, 2020. COVID-19 spread rapidly across the world and, on February 3rd, 2020, the Brazilian Ministry of Health declared a Public Health Emergency of National Importance due to human infection [2]. The situation evolves in a continuous overload in health services with the expansion of the pandemic in the country, totaling 10.517.232 cases and 254.221 confirmed deaths on 28/02/2021 [3].

COVID-19 infection may be asymptomatic or progress to Severe Acute Respiratory Syndrome (SARS-Cov-2), which requires intensive care. The main symptoms include self-reported fever, fatigue, loss of smell and taste, dry cough, myalgia and dyspnea and uncommon symptoms include sputum production, headache, hemoptysis and diarrhea [4–8].

In Brazil, primary health care (PHC) services have played an essential role in the management of COVID-19 cases. PHC main actions include measures of clinical screening, social support and reception, monitoring of home isolation until discharge from isolation. For severe cases, it includes clinical stabilization, referral and transportation to specialized centers or hospital services [9]. The PHC community-based profile allows an intimate relationship with the population under its responsibility, which can be directly related to the capacity to refrain contagious cases of COVID-19 in the Brazilian scenario from spreading.

Contagion control generally comes with quarantining of those already infected associated with other preventive measures, such as social isolation through the closing of schools, travel restrictions, use of masks, greater care in hand hygiene and objects [10–13]. Additionally, if no control measures were taken, the virus could reach 40–70% of the population, and between 5–10% of this population would need to be hospitalized [10]. As a result, social isolation, with adherence by the population, is considered a factor of great impact against the spread of the epidemic [12].

Brazil, as a country of continental dimensions, with heterogeneous social scenarios, demonstrates that the pandemic effects are also different in space and time, thus justifying analysis and different actions for the management of contagion by COVID-19 in different areas. Most of the population depends exclusively on the Unified Health System (SUS), where 72% of its locations have an insufficient amount of coverage in the number of hospital beds, mainly in intensive care [14]. In this context, the PHC role becomes even more essential, in order to guarantee the integrality of care, longitudinality of actions and coordination of care for infected cases, since it is PHC that holds the knowledge of the reality and data of each individual and population [9].

Several factors can be barriers or facilitators for a desirable outcome of COVID-19 cases in Brazil. Social, behavioral, demographic, biological and health service organization aspects can make a difference in the design and implementation of public policies to combat the pandemic [15]. Assessing these aspects in regions of social vulnerability might bring important benefits

in the quality of care for the infected, in the optimization of resources and in slowing the expansion of cases and deaths.

The factors mentioned above may have even greater impact in the Northeastern Brazil. According to the report of the United Nations Development Programme [16], this region of Brazil, despite having great natural and cultural wealth, is characterized by high levels of social inequality and income concentration, which reflects in lower education levels, quality of life and access to health and sanitation services. This profile, that may be observed in other developing countries, places the Northeastern Brazil in a situation of great social vulnerability. This context makes fighting the epidemic in this area more challenging for public managers.

Given the above, this study aims to identify the effect of PHC coverage and social isolation on the dissemination of cases and deaths by COVID-19 in Northeastern Brazil, controlling socio-demographic and economic conditions as well as the organization of health services. We hypothesize that both social isolation and PHC coverage have their effects modulated by contextual conditions in mitigating the COVID-19 pandemic.

## Materials and methods

This is an ecological time-series study with descriptive and analytical components, based on population, using secondary data. This design allows the establishment of a cause-effect relationship for public health interventions. The population from this study were residents of the main larger cities from the nine Federative Units of northeastern Brazil, in view of the greater concentration of cases in these cities and because they are generally a reference for economy and health services for other cities within the Federative Units.

Brazil is the largest country in Latin America, and it is geographically divided into five regions. The Northeastern region is historically marked by strong social inequality, with a population of approximately 53 million people. It is the region with the largest number of units of the Brazilian federation, nine in total, and with an area equivalent to around 18% of Brazilian territory [17]. In this study, we are analyzing the entire population of data, so it becomes unnecessary to have a sample size to represent the design.

Secondary data were collected from SUS Data Department (DATASUS), including the study outcomes: confirmed cases and deaths by COVID-19 [3] and some independent variables. Other data were collected in the database of the Brazilian Institute of Geography and Statistics (IBGE), which was compiled by the Brazilian agency of the United Nations Development Programme (UNDP) [16].

The time series data refer to the period from February 26 to May 16, 2020. The study primary outcome was COVID-19 cases diagnosed (y) and secondary outcome was deaths by COVID-19 (y'). The independent variables (x) were time, analyzed through epidemiological weeks; PHC coverage, Family Health Strategy (FHS) coverage through the proportion of ESF (Family Health Strategy) services in PHC in March 2020; evaluation score of the PHC health services quality by the Quality Monitoring and Evaluation Program (PMAQ); number of beds for hospitalization in April 2020; the Human Development Index (HDI), the Gini index, Gross Domestic Product (GDP) per capita, distance to the epicenter of the pandemic in Brazil, Demographic Density, percentage of employed persons and the percentage of social isolation in the nine Federative Units.

PHC coverage was stratified into less than 50%, from 50 to 74% and above 75% of the population to estimate nested relationships with other variables [18]. PMAQ score was also stratified below (<2) and above the average points (≧ 2) of the assessment. The number of hospital beds was transformed by the number of beds per thousand inhabitants (beds/population ratio) and stratified into less than 3.2 beds per thousand inhabitants and 3.2 or more beds, based on

the world average [19]. The categorization of this variable was done through the analysis of their respective quants, considering that there is no criterion available in the literature to classify the strata. The other variables were treated as they were collected.

HDI is an important tool for assessing the development of certain localities and being measured by the geometric mean of the sum of life expectancy at birth, education index and income index. In Brazil, it is used as a key index of the United Nations Millennium Development Goals [16]. The Gini index was used as an instrument to measure the degree of income concentration in the capitals, being used as a measure of social inequality, ranging from 0 to 1, and the closer to 1 the greater the inequality in that location, their categorization was done according to the distribution of their data, considering that there is no criterion for stratification available in the literature. GDP per capita was used to assess the degree of cities development, the higher its value, the more developed the city is. The distance to the epicenter of the pandemic in Brazil is measured for kilometer to São Paulo City. Demographic density was observed to portray the spatial distribution of inhabitants per square kilometer (km2) and the percentage of employed people were considered to be persons who worked for at least one full hour with remuneration in cash, products, goods or benefits (housing, food, clothing, training, etc.), or in work without direct remuneration in support of the economic activity of a member of the household or a relative who lives in another domicile, or, still, those who had paid work from which they were temporarily away that week [16].

The social isolation index was extracted from a monitoring system (https://mapabrasileirodacovid.inloco.com.br/pt/), which calculates the percentage of the population of the cities that are following isolation recommendations. It is estimated by the triangulation of cell towers to measure the displacement greater than 200 meters from the devices for personal use as well as the concentration of these people in certain locations. To this end, they use the polygons of all regions from IBGE database in order to guarantee an accurate and true to categorization. Distance to the epicenter of the pandemic in Brazil variable was include in the model for cases COVID-19 outcome and beds/population ratio was applied in death's model because conceptual reasons in relationship to outcomes.

The theoretical model analysis was set up to observe the association between the temporal evolution of COVID-19 cases and deaths. Such independent variables underwent crude analysis in order to estimate their relationship with the outcomes and then be included in an adjusted model to extract the main effects of each factor. Categorical variables also performed a nested analysis based on their conceptual and spatial relationship if they had a main effect on the adjusted model.

Statistical analysis was performed using the Generalized Linear Models (GLM) with a work correlation matrix Autoregressive (AR) of order 1, since it is a time-series where previous cases affect the number of subsequent cases, that is, there is data correlation. The log link function with Gamma distribution was used because the data of the independent variables did not present a direct linear relationship with the outcome and this was a count variable. The fit quality of the model was assessed by the quality information criterion (QIC), when the lowest estimate indicates the best fit, obeying the parsimony of the theoretical model.

Hypothesis tests were performed with the Wald chi-square test between the outcomes and the independent variables, selecting those with "p" values equal to or less than 0.10 to be included in the adjusted model. A significance level of 5% ($\alpha \leqq 0.05$) was adopted. The measure of effect used to present the relationship magnitude was the equation coefficients of the crude and adjusted model, aided by its exponentialized version. The null hypothesis of the present study is that there is no association between the outcome and the independent variables.

**Table 1. Frequency distribution of the contextual characteristics of Northeastern cities in Brazil.**

| | Total | Aracaju | Fortaleza | João Pessoa | Maceió | Natal | Recife | Salvador | São Luís | Teresina |
|---|---|---|---|---|---|---|---|---|---|---|
| COVID-19 cases | 43987 | 2032 | 15162 | 1540 | 2542 | 1288 | 9413 | 5267 | 5597 | 1146 |
| COVID-19 deaths | 2588 | 20 | 1163 | 72 | 131 | 36 | 573 | 185 | 379 | 29 |
| Lethality | 4.2 | 1.0 | 8.0 | 5.0 | 5.0 | 3.0 | 6.0 | 4.0 | 7.0 | 3.0 |
| FHS coverage (%) | 55.69 (±23.68) | 70.36 | 49.89 | 86.57 | 26.95 | 37.42 | 56.39 | 36.39 | 37.26 | 100.00 |
| PHC coverage (%) | 65.62 (±19.83) | 75.54 | 61.36 | 96.34 | 44.60 | 54.93 | 64.87 | 47.53 | 45.43 | 100.00 |
| FHS-PHC ratio (%) | 82.04 (±11.66) | 93.14 | 81.30 | 89.85 | 60.42 | 68.12 | 86.92 | 76.56 | 82.01 | 100.00 |
| PHC quality | 2.12 (±0.54) | 2.27 | 1.25 | 2.53 | 2.07 | 2.23 | 2.13 | 1.40 | 2.01 | 3.20 |
| GDP per capta | 24814.72 (±3169.57) | 25185.55 | 23436.66 | 24319.82 | 21210.09 | 26497.08 | 31743.72 | 21231.48 | 27226.41 | 22481.67 |
| Demographic Density | 3745.26 (±2330.05) | 3140.65 | 7786.44 | 3421.28 | 1854.10 | 4805.24 | 7039.64 | 3859.44 | 1215.69 | 584.94 |
| Employed persons (%) | 34.30 (±4.79) | 35.80 | 32.00 | 36.10 | 26.40 | 36.60 | 43.90 | 28.60 | 33.40 | 35.90 |
| GINI | 0.50 (±0.01) | 0.47 | 0.51 | 0.50 | 0.52 | 0.53 | 0.49 | 0.49 | 0.49 | 0.50 |
| HDI | 0.75 (±0.01) | 0.77 | 0.75 | 0.76 | 0.72 | 0.76 | 0.77 | 0.759 | 0.76 | 0.75 |
| Number of beds | 5062.55 (±2762.23) | 2382 | 8710 | 2922 | 3029 | 3138 | 9204 | 8852 | 3961 | 3365 |
| Beds/population ratio | 3.68 (±0.72) | 3.62 | 3.26 | 3.61 | 2.97 | 3.54 | 5.59 | 3.08 | 3.59 | 3.89 |
| Social Isolation (%) | 46.32 (±4.27) | 46.58 | 49.71 | 45.07 | 45.23 | 43.85 | 49.14 | 44.74 | 46.68 | 45.88 |

Coronavirus disease 2019 (COVID-19); Family Health Strategy (FHS); Primary Health Care (PHC); Gross Domestic Product (GDP); Human Development Index (HDI); Quality Monitoring and Evaluation Program (PMAQ).

## Results

In the nine cities analyzed, we identified an accumulated of 43,969 cases and 2,588 deaths due to COVID-19 up to the twentieth epidemiological week. The average number of diagnosed in the cities evaluated cases was 4885.44 (± 1574.85), 287.55 (± 125.80) of deaths and an overall lethality of 4.4%.

PHC has an average population coverage of 65.62% in the sample, but with great variability (±19.83) between cities as well as its quality, which has an average of 2.12, with very different outliers (Table 1). GDP per capita (24814.72±3169.57), demographic density (3745.26 ±2330.05) and the beds/population ratio (3.68±0.72), also revealed a diversity of scenarios.

Social isolation proves to be low in the sample studied, 46.32 (± 4.27), compared to what is recommended by the scientific and health authorities. The characterization by income can be seen in Table 1.

The epidemic evolution of COVID-19 cases in the sample reveals an exponential curve pattern, following the equation $y = 529.66^{0.72x}$, which explains 89% of data variability ($R2_{ajus}$ = 0.89). Death events demonstrate an evolution of the exponential curve, $y = 7.90^{0.94x}$, which explains 87% of cases variability ($R2_{ajus}$ = 0.87). It is possible to identify "front steps" pattern in the occurrence of COVID-19 cases in the analyzed capitals, due to the magnitude of the dissemination of cases. The same is observed for cases of death (Fig 1).

In Table 2, crude analysis for COVID-19 new cases shows that the temporality of epidemiological weeks ($\chi 2$ = 541.76; degree freedom (df) = 1; *p-value*<0.001), PHC coverage strata ($\chi 2$ = 40.78; df = 2; *p-value*<0.001), PHC quality strata ($\chi 2$ = 48.79; df = 1; *p-value*<0.001), demographic density ($\chi 2$ = 5.64; df = 1; *p-value* = 0.01) and social isolation rate ($\chi 2$ = 7.96; df = 1; *p-value* = 0.005) establish an association with the viral outbreak. As for the cases of deaths due to COVID-19, the association is observed with temporality of the epidemiological weeks ($\chi 2$ = 2599.84; df = 1; *p-value*<0.001), PHC coverage strata ($\chi 2$ = 24.85; df = 2; *p-value*<0.001), PHC quality strata ($\chi 2$ = 55.42; df = 1; *p-value*<0.001) and social isolation rate ($\chi 2$ = 9.48; gl = 1; *p-value* = 0.002) as well as the demographic density ($\chi 2$ = 3.12; df = 1; *p-*

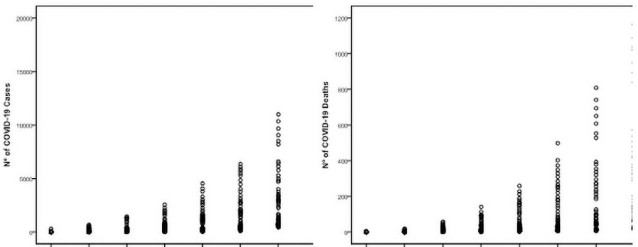

**Fig 1. Temporal distribution of COVID-19 cases and their deaths in the capitals of the Northeast of Brazil.** It shows the occurrence of distribution "front steps" motivated by speed and different magnitudes of contagion in the analyzed capitals.

value = 0.07), bed/population ratio ($\chi 2$ = 2.75; df = 1; *p-value* = 0.09) and the quantity of COVID-19 cases ($\chi 2$ = 3.10; df = 1; *p-value* = 0.07) diagnosed with marginal significance (*p-value*<0.10), Table 3.

It is relevant to highlight the non-significant effect in the crude analysis of social variables, such as Gini index and HDI, as well as the economic variables, such as GDP per capita of the cities. FHS coverage also showed no association with the time-series of COVID-19 cases (Table 2) and deaths (Table 3).

After adjusting for confounding variables, there are major effects of the pandemic temporality ($\chi 2$ = 625.38; df = 1; *p-value*<0.001), social isolation rate ($\chi 2$ = 365.99; df = 1; *p-value*<0.001), demographic density ($\chi 2$ = 3.97; df = 1; *p-value* = 0.04) and PHC coverage ($\chi 2$ = 9.27; df = 2; *p-value* = 0.01) in the number of cases accumulated in the studied sample. There is a progression over time (B = 0.17; *p-value*<0.001) that was already predicted because

**Table 2. Crude model for association with incident COVID-19 cases in the time-series of twenty epidemiological weeks in Northeastern cities in Brazil.**

| Variable | B | Hypothesis test | | |
|---|---|---|---|---|
| | | Wald $\chi 2$ | df | *p-value* |
| Epidemiological week | 0.12 | 541.76 | 1 | <0.001 |
| PHC coverage | | | | |
| <50% | 1.12 | 25.99 | 1 | <0.001 |
| 50–74% | 1.80 | 20.29 | 1 | <0.001 |
| ≥75% | 0 | | | |
| FHS-PHC ratio | -0.006 | 0.09 | 1 | 0.76 |
| PMAQ assessment | | | | |
| Below average | 1.15 | 5.68 | 1 | 0.017 |
| Above average | 0 | | | |
| GDP per capta | 5.264E-5 | 0.86 | 1 | 0.35 |
| Demographic Density | 2.28E-4 | 5.65 | 1 | 0.01 |
| Employed persons (%) | 0.008 | 0.02 | 1 | 0.86 |
| HDI | 9.42 | 0.85 | 1 | 0.35 |
| GINI | -1.49 | 0.01 | 1 | 0.92 |
| Social Isolation (%) | .001 | 7.96 | 1 | 0.005 |
| Epicenter Distance | 0.001 | 0.81 | 1 | 0.36 |

Family Health Strategy (FHS); Primary Health Care (PHC); Gross Domestic Product (GDP); Human Development Index (HDI); Quality Monitoring and Evaluation Program (PMAQ). Model regression coefficient (B); Degrees of Freedom (df).

**Table 3. Crude model for association with COVID-19 deaths in the time-series of twenty epidemiological weeks in Northeastern cities in Brazil.**

| Variable | B | Hypothesis test | | |
|---|---|---|---|---|
| | | Wald χ2 | df | *p-value* |
| Epidemiological week | 0.12 | 2599.84 | 1 | <0.001 |
| COVID-19 cases | 2.05E-4 | 3.10 | 1 | 0.07 |
| PHC coverage | | | | |
| <50% | 1.62 | 13.85 | 1 | <0.001 |
| 50–74% | 2.56 | 21.84 | 1 | <0.001 |
| ≥75% | 0 | | | |
| FHS-PHC ratio | -0.006 | 0.068 | 1 | 0.794 |
| PMAQ assessment | 1.28 | 3.78 | 1 | 0.05 |
| Below average | | | | |
| Above average | 0 | | | |
| GDP per capta | 7.144E-5 | 1.17 | 1 | 0.27 |
| Demographic Density | 2.58E-4 | 3.12 | 1 | 0.07 |
| Employed persons (%) | 0.01 | 0.05 | 1 | 0.80 |
| GINI | 8.69 | 0.20 | 1 | 0.65 |
| HDI | 10.00 | 0.57 | 1 | 0.44 |
| Social Isolation (%) | 0.001 | 9.48 | 1 | 0.002 |
| Beds/population ratio | | | | |
| <3.2 | -0.77 | 2.75 | 1 | 0.09 |
| ≥3.2 | 0 | | | |

Coronavirus disease 2019 (COVID-19); Family Health Strategy (FHS); Primary Health Care (PHC); Gross Domestic Product (GDP); Human Development Index (HDI); Quality Monitoring and Evaluation Program (PMAQ). Model regression coefficient (B); Degrees of Freedom (df).

it is a pandemic and that slightly increases in contexts of greater population density (B = 2.46E-4; *p-value*<0.001). On the other hand, social isolation rate shows a negative association with the number of cases (B = -0.007; *p-value*<0.001) as well as PHC coverage above 50%, where cities with coverage below 50% present on average three times more cases than those with coverage greater than 75%. This seems to be independent of the quality of PHC assistance (Table 4).

For deaths, there are major effects of pandemic temporality (χ2 = 70.87; df = 1; *p-value*<0.001), social isolation rate (χ2 = 41.04; df = 1; *p-value*<0.001), demographic density (χ2 = 4.78; df = 1; *p-value* = 0.02), COVID-19 cases (χ2 = 4.94; df = 1; *p-value* = 0.02), PHC coverage (χ2 = 31.75; df = 2; *p-value*<0.001) and bed/population ratio (χ2 = 9.02; df = 1; *p-value* = 0.003). Pandemic temporality (B = 0.07), more slightly the demographic density (B = 3.20E-4) and the number of diagnosed COVID-19 cases (B = 1.62E-4) were related to the increase in deaths from the virus. In contrast, social isolation rate (B = -0.004) demonstrates an inversely proportional relationship, as well as the nesting of PHC coverage variables and beds/population ratio (χ2 = 30,87; df = 3 *p-value*<0.001), where we identified that PHC coverage strata above 50% together with the supply of hospital beds above 3.2 per thousand inhabitants has two (B = 0.85) to six times (B = 1.89) less chances of having deaths due to COVID-19 (Table 5).

## Discussion

Seeking to test the theoretical hypothesis of the study to estimate the effect of PHC and social isolation on the time-series of COVID-19 cases and deaths, the analysis points out that the

**Table 4. Adjusted model for the association with incident COVID-19 cases in the time-series of twenty epidemiological weeks in Northeastern cities in Brazil.**

| Variable | B | Hypothesis test | | | RR | CI95% | |
|---|---|---|---|---|---|---|---|
| | | Wald χ2 | df | *p-value* | | Lower | Upper |
| Interception | 2.59 | 60.15 | 1 | <0.001 | 13.36 | 6.94 | 25.72 |
| Epidemiological week | 0.17 | 625.38 | 1 | <0.001 | 1.19 | 1.18 | 1.21 |
| Social Isolation (%) | -0.007 | 365.99 | 1 | <0.001 | 0.993 | 0.992 | 0.994 |
| Demographic Density | 2.46E-4 | 3.97 | 1 | 0.04 | 1.0002 | 1.000004 | 1.0004 |
| PHC coverage | | | | | | | |
| <50% | 1.12 | 9.09 | 1 | 0.003 | 3.08 | 1.48 | 6.42 |
| 50–74% | 0.53 | 0.62 | 1 | 0.42 | 1.70 | 0.45 | 6.36 |
| ≥75% | 0 | | | | 1 | | |
| PMAQ Assessment | | | | | | | |
| Below average | 0.310 | 0.67 | 1 | 0.41 | 1.36 | 0.65 | 2.85 |
| Above average | 0 | | | | 1 | | |
| Scale | 0.72 | | | | | | |

Primary Health Care (PHC); Quality Monitoring and Evaluation Program (PMAQ). Model regression coefficient (B); Degrees of Freedom (df) Risk Relative (RR); Confidence Interval (CI).

extent of PHC coverage in the cities of Northeastern Brazil together with social distancing are associated with fewer cases and deaths up to the 20th epidemiological week. These are likely to be the modifiable factors listed by this study for timely actions by health system managers in order to mitigate the spread of the epidemic and its consequences for the health of the population [20].

The cities included in this study show an exponential growth in COVID-19 cases after 50 days of monitoring by the Ministry of Health, similar to countries like China [21,22], which may influence the use of hospital services in the same pattern as it happened in Lombardy,

**Table 5. Adjusted model for the association with COVID-19 deaths in the time-series of twenty epidemiological weeks in Northeastern cities in Brazil.**

| Variable | B | Hypothesis test | | | RR | CI95% | |
|---|---|---|---|---|---|---|---|
| | | Wald χ2 | df | *p-value* | | Lower | Upper |
| Interception | 0.76 | 3.48 | 1 | 0.06 | 2.14 | 0.96 | 4.77 |
| Epidemiological week | 0.07 | 257.64 | 1 | <0.001 | 1.07 | 1.06 | 1.08 |
| COVID-19 cases | 1.62E-4 | 4.62 | 1 | 0.03 | 1.0001 | 1.00001 | 1.0003 |
| Demographic Density | 3.20E-4 | 4.63 | 1 | 0.03 | 1.0003 | 1.00002 | 1.001 |
| Social Isolation (%) | -0.004 | 41.04 | 1 | <0.001 | 0.996 | 0.995 | 0.997 |
| PMAQ Assessment | | | | | | | |
| Below average | -0.26 | 0.44 | 1 | 0.50 | 0.76 | 0.34 | 1.69 |
| Above average | 0 | | | | 1 | | |
| PHC coverage (Beds/population ratio) | | | | | | | |
| <50% PHC with <3.2 beds/pop. | 0.85 | 7.83 | 1 | 0.005 | 2.35 | 1.29 | 4.29 |
| <50% PHC with ≥3.2 beds/pop. | 1.89 | 30.55 | 1 | <0.001 | 6.64 | 3.39 | 13.01 |
| 50–74% PHC with ≥3.2 beds/pop. | -0.26 | 0.14 | 1 | 0.70 | 0.76 | 0.19 | 2.98 |
| ≥75% PHC with ≥3.2 beds/pop | 0 | | | | 1 | | |
| Scale | 0.74 | | | | | | |

Coronavirus disease 2019 (COVID-19); Primary Health Care (PHC); Quality Monitoring and Evaluation Program (PMAQ). Model regression coefficient (B); Degrees of Freedom (df) Risk Relative (RR); Confidence Interval (CI).

Italy [23]. However, the pattern is not homogeneous across all capitals, which leads us to assume that there are contextual factors that may be modulating this dissemination and also deaths.

Lethality is lower than Italy with a month of outbreak and higher than China for the same period [24], but dissemination and deaths significantly vary in cities such as Fortaleza and Aracajú. This indicator is extremely influenced by the number of cases, population demographic factors and, mainly, by the health system's capacity to cope, specifically the installed capacity of intensive care services, material resources such as mechanical respirators and trained human resources [24].

PHC and access to it are essential for the progress of a population's health conditions, including communicable diseases. One of SUS principles is universal access, a very important step towards achieving improvements in health indicators, however not all the population is effectively covered by the PHC system in Brazil [17]. In this study, there was a significant association between low PHC coverage with cases and deaths from COVID-19, controlling the effects of social isolation, demographic density and serial evolution. It can be inferred that infection by COVID-19 is a PHC-sensitive condition for mitigate the pressure on hospital services that are limited mainly in socioeconomic vulnerable regions.

To date, no other studies have addressed these topics. Greater PHC coverage promotes a decrease in new cases, hospitalizations and deaths, as it has already been shown to be effective for other health conditions such as HIV control, infant mortality, stroke, arterial hypertension and diabetes [25,26]. The northeastern region of Brazil is the one that has the largest current coverage and the greatest temporal evolution of it [17,25]. However, we have no way of guessing how long PHC will endure this burden and also fail to provide care for the other health conditions in which it has been successful.

The likely explanation for the mitigating effect of PHC on the spread of COVID-19 is in the portfolio of its attributions, functions and micropolitics. In summary: it is community-based and linked to the population's territory, it develops capillary actions for stratification of social and biological vulnerability, screening of health conditions, individual and collective monitoring of cases, and coordination of care within the network [27]. These attributes and functions can enhance specific actions for the moment, such as educating users about social isolation, monitoring suspected and confirmed cases, as well as serving as a link to other types of social support, such as access to emergency government aid [9].

In addition, we would also like to highlight that this study observed an inverse association between social isolation and COVID-19 cases and deaths. Isolation is one of the main measures used to control these cases. Corroborating our findings, the study by Ferguson et al. [28], also known as the Imperial College Study, showed that combining quarantine together with social isolation were effective in controlling the spread of COVID-19. A systematic review published in the Cochrane Library database [11] evaluated the effect of quarantine alone and combined with social distance, showing that associated measures seem to be more effective.

Adherence to social isolation in the studied population was considered low. Even so, we had an inverse and significant correlation between social isolation and the diagnosed cases and deaths by COVID-19. However, it is likely that greater population adherence to social isolation would lead to stronger effects in reducing contagion. Thus, for this moment, it is recommended that isolation measures be carried out or maintained combined with school closures, travel restrictions, social distance and others, from the initial stage of the outbreak. Social restriction measures, despite appearing to be more effective in epicenters of the disease, also have effects in cities with fewer cases, minimizing the spread of the epidemic [11,22,28,29].

As COVID-19 is mainly transmitted through the airways and respiratory droplets, a high population concentration in the same place would facilitate transmission. In our study, the

cities with the highest demographic density, Fortaleza and Recife, had a higher number of cases and deaths. These results are reinforced by findings from other studies [7]. Furthermore, the cities Fortaleza and Recife have been in critical situation for the management of the health system due to the greater spread of the disease and high number of deaths. This reinforces the hypothesis that it is a great challenge to preserve social distancing in an area of greater concentration of people in space. Despite the demographic density being a difficult modifiable factor, other prophylaxis actions for COVID-19 are necessary, such as: PHC reinforcement and increase in number of hospital beds, public hygiene points, distribution of personal protective equipment, among others.

Along with PHC coverage and social isolation, beds/population ratio for hospitalization above the world average of 3.2 per thousand inhabitants showed an effect in mitigating the drastic events of deaths in those infected with COVID-19 in the studied sample. This is due to the virus's ability to generate critical clinical cases in subpopulations with biological vulnerability such as diabetes, hypertension and chronic kidney disease [23]. However, the nested analysis of the beds/population ratio and PHC coverage showed the effect of the interaction between these variables, meaning that the implementation of more hospital beds for critical care without the proper support for PHC expansion may be ineffective for controlling the pandemic due to the non-stagnation of new cases. This is the example of the city of São Luís, which has a hospital beds/population ratio above the world average, but with low PHC coverage, presenting lethality similar to large cities such as Fortaleza and Recife.

In our study, social and economic indicators such as the Gini index, HDI, GDP per capita and ratio of persons employed were not significantly related to the increase in the number of cases and deaths in northeastern cities, which may indicate that transmission of COVID-19 is not influenced by these variables after the transmission started, unlike other locations in the world [30]. Thus, COVID-19 spreads in space regardless of social or economic stratification in these cities. High HDI countries, such as Spain (0.893), France (0.891), United Kingdom (0.920) and Sweden (0.937) were strongly impacted by this pandemic [16].

Despite the impactful findings for Brazilian public health, it is prudent to mention some limitations that must be analyzed. The first is that COVID-19 is underdiagnosed in Brazil as well as elsewhere in the world, which leads to underreporting of events. A second limitation is the outdating of ecological measures such as the HDI and Gini index in the Brazilian information systems, however we believe that their modification over time is minimal. The third limitation concerns the use of the social isolation rate of the federal unit as a proxy for the performed in the city and as an aggregate data of individual observation measures.

The COVID-19 pandemic brings new elements and specific characteristics that affect each region of the world in a specific way. This originality poses as a great challenge for the reorganization of health services across the planet. From the variables addressed in this study, we state that an expansion and qualification of PHC services, associated with an increase in the number of hospital beds for care, as well as maintenance policies or even expansion of social restrictions and possible relaxation of distance measures in opportune time and manner, are important for the current pandemic scenario in Northeastern Brazil. Additional COVID-19 prevention strategies, not measured in this study, are relevant tools that should be considered by public managers.

Although the Brazilian Federal Government declared Public Health Emergency of National Importance in February 2020, effective actions to prevent and combat the epidemic were taken late, contradicting the dimension of the public health problem that this condition presented to the country. Including nowadays, wrong decisions and guidelines by public authorities are part of the routine in the Brazilian Government. Well-articulated actions between public managers and political leaders, with a positioning based on scientific evidence and

technical criteria, may provide time and breath for health services to adjust to the new demands produced by this pandemic. Other regions of Brazil and the world with characteristics of social vulnerability and organization of health services similar to those of the cities of Northeastern Brazil, can make use of the results found here in assisting decision making, with regard to public policies to be adopted and also to the guidelines for the population.

## Supporting information

**S1 File.**
(XLS)

## Author Contributions

**Conceptualization:** Sanderson José Costa de Assis, Johnnatas Mikael Lopes, Marcello Barbosa Otoni Gonçalves Guedes, Geronimo José Bouzas Sanchis, Diego Neves Araujo, Angelo Giuseppe Roncalli.

**Data curation:** Sanderson José Costa de Assis, Johnnatas Mikael Lopes, Angelo Giuseppe Roncalli.

**Formal analysis:** Sanderson José Costa de Assis, Johnnatas Mikael Lopes, Angelo Giuseppe Roncalli.

**Investigation:** Johnnatas Mikael Lopes.

**Methodology:** Sanderson José Costa de Assis, Johnnatas Mikael Lopes, Marcello Barbosa Otoni Gonçalves Guedes, Geronimo José Bouzas Sanchis, Diego Neves Araujo.

**Writing – review & editing:** Sanderson José Costa de Assis, Johnnatas Mikael Lopes, Marcello Barbosa Otoni Gonçalves Guedes, Geronimo José Bouzas Sanchis, Diego Neves Araujo, Angelo Giuseppe Roncalli.

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
