## [Decision Letter · Decision Letter 0]

8 Feb 2021

PONE-D-20-19401

PRIMARY HEALTH CARE AND SOCIAL ISOLATION AGAINST COVID-19 IN NORTHEASTERN BRAZIL: ECOLOGICAL TIME-SERIES STUDY

PLOS ONE

Dear Dr. Assis,

Thank you for submitting your manuscript to PLOS ONE. After careful consideration, we feel that it has merit but does not fully meet PLOS ONE’s publication criteria as it currently stands. Therefore, we invite you to submit a revised version of the manuscript that addresses the points raised during the review process, by the four reviewer.

We look forward to receiving your revised manuscript.

Kind regards,

Lidia Adriana Braunstein, PhD

Academic Editor

PLOS ONE

Journal Requirements:

Could you therefore please include the title page into the beginning of your manuscript file itself, listing all authors and affiliations

3.We suggest you thoroughly copyedit your manuscript for language usage, spelling, and grammar. If you do not know anyone who can help you do this, you may wish to consider employing a professional scientific editing service.  

4. Please include a copy of Table 4 and 5 which you refer to in your text on page 4.

Reviewers' comments:

Reviewer's Responses to Questions

**Comments to the Author**

1. Is the manuscript technically sound, and do the data support the conclusions?

Reviewer #1: Yes

Reviewer #2: Yes

Reviewer #3: Yes

Reviewer #4: Yes

2. Has the statistical analysis been performed appropriately and rigorously? 

Reviewer #1: Yes

Reviewer #2: Yes

Reviewer #3: Yes

Reviewer #4: Yes

3. Have the authors made all data underlying the findings in their manuscript fully available?

Reviewer #1: Yes

Reviewer #2: Yes

Reviewer #3: Yes

Reviewer #4: Yes

4. Is the manuscript presented in an intelligible fashion and written in standard English?

Reviewer #1: Yes

Reviewer #2: Yes

Reviewer #3: Yes

Reviewer #4: Yes

5. Review Comments to the Author

Reviewer #1: The study presents the results of an original research, important for a famous and known region as the Northeast of Brazil.

The analyzes were carried out to a high technical standard and are described in sufficient detail.

The conclusions were presented in an appropriate manner and supported by the data.

I would like to mention in the text that some observational studies checklist (STROBE?) was followed.

Reviewer #2: This is a relevant study, which shows the concern and influence of primary health care against Pandemic by COVID-19.

Minor revisions are suggested.

In the Background:

Update of epidemiological data to date, although the study has been based on a previous situation.

Add the characteristic symptoms of loss of smell and taste, in some cases, as already well evidenced in the literature.

In the discussion

The paragraphs are long and some information is repeated, which can be reduced.

Final considerations

Just one paragraph can be enough to complete the study. I suggest summarizing the last one or putting it in the previous section as suggestions.

Reviewer #3: Dear Author,

The article is interesting and contributes a lot to the performance of health services in Brazil and in countries with similar characteristics in the particular context of coronavirus disease (COVID-19), but requires some revisions. The study brings as strengths the fact that the infection by COVID-19 is a condition sensitive to PHC, which would reduce the demand for hospital services, in addition to the inclusion of social and economic variables in the analysis of the number of cases and deaths of the population studied and as a weak point the lack of access to individual data, characteristic of the type of study.

In the abstract on page 1, line 9, I suggest reviewing the information on "Brazil is the new epicenter of coronavirus disease", due to the constant changes in the pandemic scenario.

In Introduction, line 45 on page 3, I suggest updating the number of cases and deaths due to their rapid growth and the importance of this information in this context.

In the materials and methods, in line 115 on page 6, I suggest placing the start date of the time-series. Line 127 on page 6 was missing the reference to stratify the PMAQ score. Line 138 on page 7 was missing the reference to categorize the Gini index.

In the results, on lines 225 and 237, tables 4 and 5 are not present in the submission file, I suggest inserting the tables or removing this information if it does not exist.

Table 1: Authors should review the sum of the frequencies of the number of cases compared to the total value presented in the same table.

In the discussion, line 268, the authors state that greater PHC coverage reduces the number of cases of hospitalization and deaths, as it has not proved to be effective for other health conditions.

This statement seems contradictory within the context already exposed by the authors regarding PHC in the manuscript and in the finalization of the paragraph of that statement. I suggest revising the translation of this sentence or making it clearer how PHC coverage would influence the reduction of cases and deaths due to COVID-19 and not in other situations.

Line 301: I suggest making it clear what is considered a suitable environment for the transmission of COVID-19

Reviewer #4: In this manuscript, the authors evaluate the possible correlation between positive cases and deaths because of COVID-19 and various demographic indexes, among which primary health care coverage stands out. I find this work very interesting, pointing out several key points to take note of in order to face the current pandemic, as well as the possible future challenges that public health could face. I have, however, some comments that I would like the authors to address:

Main points:

1) In line 194, the authors point out that the death curve follows a cubic function, a choice that generates some doubts. The R-squared value of 0.23 does not seem to be very encouraging, but above all, I find it difficult to justify from an epidemic theory a curve of the cubic type, with a local minimum close to zero at x approximately 14. Can the authors justify this choice? In addition, it would be good to provide graphs where the mathematical curve and the real data are included.

2) Given the months that have passed since the initial submission, the authors may wish to update the results presented. I also understand that the authors might prefer to focus only on the initial phase of exponential case growth. Whatever the choice, I would suggest clarifying it in the text and reviewing comments that the authors might have made about the future (for example, line 30).

Minor points:

3) In general, the text is written in good English. Although I would check the word "contamination" (line 12), "tax" (line 23), "persons" (line 142).

4) When writing the models with the mathematical equation y(x), the authors should make it explicit in the text what represents each variable.

5) Mathematical variables (x, y, p, chi^2, etc.) should be written in italics, this should be checked throughout the manuscript. Do not use "p" (line 173). but p-value.

6) The symbol that corresponds to the Chi-square value must be the Greek symbol, not a Latin x since it is confused with the variable x in mathematical models. Please, check the entire manuscript.

7) On line 25, a parenthesis is missing and there is a double ";" when the Chi-square value and the p-value are reported. On line 144, there is an extra space after the word “clothing”.

8) I find some inconsistencies in the use of upper and lower case throughout the text. Please, check:

- line 28: covid

- lines 99 and 103: federative units of northeastern

- line 142: Km

- lines 185, 190, 198: table

9) In the paragraph on line 172, or where the authors deem appropriate, the null hypothesis of the analysis should be explicitly clarified. Although it is standard that the null hypothesis is that the variables are independent, this is an arbitrary choice, so it should be clarified in the text.

10) In lines 225 and 237, reference is made to tables 4 and 5, which do not exist.

11) I find some drawbacks with the paragraph between lines 96 and 101:

- Line 96: what is "it"? It is preferable to use "in this work / study ...".

- Line 97: what is "secondary data"?

- In the last sentence, the first half (lines 98-99) is repeated with the next paragraph (line 102).

- Review the wording of the phrase "to be commonly an economic reference and health organization for cities ...".

- In general I find this paragraph somewhat weak compared to the paragraphs that follow. Authors could evaluate a general edition of this paragraph.

12) On line 106, I suggest a period instead of a comma after “million people”.

13) I find some drawbacks in the sentence on line 110. The beginning is redundant (“were collected” is mentioned twice). I don't understand what “(3)” represents. I suggest clarifying what the "other independent variables" are.

14) Line 119. What is ESF?

15) Line 180. Is the average taken between cities? Or maybe days?

In summary, I think the authors present a good work, which could benefit from the points I make. I recommend a minor review of the work.

6. PLOS authors have the option to publish the peer review history of their article (what does this mean?). If published, this will include your full peer review and any attached files.

Reviewer #1: **Yes: **Marcelo Cardoso de Souza

Reviewer #2: No

Reviewer #3: **Yes: **Thais Sousa Rodrigues Guedes

Reviewer #4: No

---

## [Author Response · Author response to Decision Letter 0]

25 Mar 2021

Lidia Adriana Braunstein

Editor-in-Chief

Plos One

 march 04, 2021.

Dear Lidia Adriana Braunstein,

 Thank you for your email with the reviewers’ comments. We have reviewed the comments and edited the manuscript accordingly. Please, find attached our point-by-point response to the reviewers. All authors have read this protocol and agree with the Plos One policy. We hope the revised manuscript is now suitable for publication. 

Sincerely. Sanderson José Costa de Assis.

Reviewer Comments:

Reviewer #1

1. The study presents the results of an original research, important for a famous and known region as the Northeast of Brazil.

The analyzes were carried out to a high technical standard and are described in sufficient detail.

The conclusions were presented in an appropriate manner and supported by the data.

I would like to mention in the text that some observational studies checklist (STROBE?) was followed.

Response: Thank you for your comments.

Reviewer #2: This is a relevant study, which shows the concern and influence of primary health care against Pandemic by COVID-19.

Background

1. Update of epidemiological data to date, although the study has been based on a previous situation.

Response: Thank you for your comments. The sentence was rewritten:

The situation evolves in a continuous overload in health services with the expansion of the pandemic in the country, totaling 10.517.232 cases and 254.221 confirmed deaths on 28/02/2021

2. Add the characteristic symptoms of loss of smell and taste, in some cases, as already well evidenced in the literature.

Response: Thank you for your comments. The sentence was rewritten:

The main symptoms include self-reported fever, fatigue, loss of smell and taste, dry cough, myalgia and dyspnea and uncommon symptoms include sputum production, headache, hemoptysis and diarrhea.

Discussion

1. The paragraphs are long and some information is repeated, which can be reduced.

Response: Thank you for your comments. Changes were made to the text

Final considerations

1. Just one paragraph can be enough to complete the study. I suggest summarizing the last one or putting it in the previous section as suggestions.

Response: Thank you for your comments. The sentence was rewritten:

Although the Brazilian Federal Government declared Public Health Emergency of National Importance in February 2020, effective actions to prevent and combat the epidemic were taken late, contradicting the dimension of the public health problem that this condition presented to the country. Including nowadays, wrong decisions and guidelines by public authorities are part of the routine in the Brazilian Government. Well-articulated actions between public managers and political leaders, with a positioning based on scientific evidence and technical criteria, may provide time and breath for health services to adjust to the new demands produced by this pandemic. Other regions of Brazil and the world with characteristics of social vulnerability and organization of health services similar to those of the cities of Northeastern Brazil, can make use of the results found here in assisting decision making, with regard to public policies to be adopted and also to the guidelines for the population.

Reviewer #3: The article is interesting and contributes a lot to the performance of health services in Brazil and in countries with similar characteristics in the particular context of coronavirus disease (COVID-19), but requires some revisions. The study brings as strengths the fact that the infection by COVID-19 is a condition sensitive to PHC, which would reduce the demand for hospital services, in addition to the inclusion of social and economic variables in the analysis of the number of cases and deaths of the population studied and as a weak point the lack of access to individual data, characteristic of the type of study.

Abstract 

1. In the abstract on page 1, line 9, I suggest reviewing the information on "Brazil is the new epicenter of coronavirus disease", due to the constant changes in the pandemic scenario.

Response: Thank you for your comments. The sentence was rewritten:

Brazil is witnessing a massive increase of corona virus disease (COVID-19).

Introduction

1. In Introduction, line 45 on page 3, I suggest updating the number of cases and deaths due to their rapid growth and the importance of this information in this context.

Response: Thank you for your comments. The sentence was rewritten:

The situation evolves in a continuous overload in health services with the expansion of the pandemic in the country, totaling 10.517.232 cases and 254.221 confirmed deaths on 28/02/2021

Materials e methods

1. In the materials and methods, in line 115 on page 6, I suggest placing the start date of the time-series.

Response: Thank you for your comments. The sentence was rewritten:

The time series data refer to the period from February 26 to May 16, 2020

2. Line 127 on page 6 was missing the reference to stratify the PMAQ score.

Response: Thank you for your comments. The sentence was rewritten:

The categorization of this variable was done through the analysis of their respective quants, considering that there is no criterion available in the literature to classify the strata.

3. Line 138 on page 7 was missing the reference to categorize the Gini index.

Response: Thank you for your comments. The sentence was rewritten:

Their categorization was done according to the distribution of their data, considering that there is no criterion for stratification available in the literature.

Results

1. In the results, on lines 225 and 237, tables 4 and 5 are not present in the submission file, I suggest inserting the tables or removing this information if it does not exist.

Response: Thank you for your comments. The tables were inserted.

2. Table 1: Authors should review the sum of the frequencies of the number of cases compared to the total value presented in the same table.

Response: Thank you for your comments. Correction was performed.

Discussion

1. In the discussion, line 268, the authors state that greater PHC coverage reduces the number of cases of hospitalization and deaths, as it has not proved to be effective for other health conditions.

This statement seems contradictory within the context already exposed by the authors regarding PHC in the manuscript and in the finalization of the paragraph of that statement. I suggest revising the translation of this sentence or making it clearer how PHC coverage would influence the reduction of cases and deaths due to COVID-19 and not in other situations.

Response: Thank you for your comment. The sentence “it has not been shown to be effective” was rewritten as “it has already been shown to be effective”.

2. Line 301: I suggest making it clear what is considered a suitable environment for the transmission of COVID-19.

Response: Thank you for your comments. The sentence was rewritten:

As COVID-19 is mainly transmitted through the airways and respiratory droplets, a high population concentration in the same place would facilitate transmission.

Reviewer #4: In this manuscript, the authors evaluate the possible correlation between positive cases and deaths because of COVID-19 and various demographic indexes, among which primary health care coverage stands out. I find this work very interesting, pointing out several key points to take note of in order to face the current pandemic, as well as the possible future challenges that public health could face. I have, however, some comments that I would like the authors to address:

1. In line 194, the authors point out that the death curve follows a cubic function, a choice that generates some doubts. The R-squared value of 0.23 does not seem to be very encouraging, but above all, I find it difficult to justify from an epidemic theory a curve of the cubic type, with a local minimum close to zero at x approximately 14. Can the authors justify this choice? In addition, it would be good to provide graphs where the mathematical curve and the real data are included.

Response: 

We are grateful for the relevant observation of the reviewer and found a mistake in the description of the curve equation. Both case and death distributions have an exponential curve. In addition, two graphs with case distributions are attached to the body of the results.

2. Given the months that have passed since the initial submission, the authors may wish to update the results presented. I also understand that the authors might prefer to focus only on the initial phase of exponential case growth. Whatever the choice, I would suggest clarifying it in the text and reviewing comments that the authors might have made about the future (for example, line 30).

Response: Thank you for your comments. We prefer to focus only on the initial phase of exponential case growth.

3. In general, the text is written in good English. Although I would check the word "contamination" (line 12), "tax" (line 23), "persons" (line 142).

Response: Thank you for your comment. The words were replaced by “confirmed cases”, “rate”, and “people”.

4. When writing the models with the mathematical equation y(x), the authors should make it explicit in the text what represents each variable.

Response: 

We appreciate the question. The algebraic representations of the model were specified in the method.

5. Mathematical variables (x, y, p, chi^2, etc.) should be written in italics, this should be checked throughout the manuscript. Do not use "p" (line 173). but p-value.

Response: Thank you for your comments. Altration was performed in the entire manuscript.

6. The symbol that corresponds to the Chi-square value must be the Greek symbol, not a Latin x since it is confused with the variable x in mathematical models. Please, check the entire manuscript.

Response: Thank you for your comments. Altration was performed in the entire manuscript.

7. On line 25, a parenthesis is missing and there is a double ";" when the Chi-square value and the p-value are reported. On line 144, there is an extra space after the word “clothing”.

Response: Thank you for your comments. The sentences have been rewritten:

On line 25: Capitals with hospital beds ≥ 3.2 per thousand inhabitants had fewer deaths (χ²=9.02; p-value=0.003), but this was influenced by PHC coverage (χ²=30,87; p-value<0.001).

On line 144: Demographic density was observed to portray the spatial distribution of inhabitants per square kilometer (Km²) and the percentage of employed persons were considered to be persons who worked for at least one full hour with remuneration in cash, products, goods or benefits (housing, food, clothing, training, etc.), or in work without direct remuneration in support of the economic activity of a member of the household or a relative who lives in another domicile, or, still, those who had paid work from which they were temporarily away that week.

8. I find some inconsistencies in the use of upper and lower case throughout the text. Please, check:

- line 28: covid

- lines 99 and 103: federative units of northeastern

- line 142: Km

- lines 185, 190, 198: table

Response: Thank you for your comments. Adjustments were made in every manuscript.

9. In the paragraph on line 172, or where the authors deem appropriate, the null hypothesis of the analysis should be explicitly clarified. Although it is standard that the null hypothesis is that the variables are independent, this is an arbitrary choice, so it should be clarified in the text.

Response: Thank you for your comments. The following sentence has been added.

The null hypothesis of the present study is that there is no association between the outcome and the independent variables.

10. In lines 225 and 237, reference is made to tables 4 and 5, which do not exist.

Response: Thank you for your comments. The tables were inserted.

11. I find some drawbacks with the paragraph between lines 96 and 101:

- Line 96: what is "it"? It is preferable to use "in this work / study ...".

- Line 97: what is "secondary data"?

- In the last sentence, the first half (lines 98-99) is repeated with the next paragraph (line 102).

- Review the wording of the phrase "to be commonly an economic reference and health organization for cities ...".

- In general I find this paragraph somewhat weak compared to the paragraphs that follow. Authors could evaluate a general edition of this paragraph.

Response: Thank you for your comments. Please find the responses below topic by topic.

- Text was rewritten in line 96.

- Secondary data refers to data collected by other data sources different than the data directly collected for this specific study. The secondary data in this study was collected from the public national Brazilian database Datasus, IBGE and UNDP.

- Sentence from lines 98-99 were rewritten with information from line 102, that was completed changed to avoid repetition.

- The sentence was reviewed and rewritten for better understanding.

- We believe that the editions after the reviewer’s comments and suggestions have improved the paragraph.

12. On line 106, I suggest a period instead of a comma after “million people”.

Response: Thank you for your comments. The sentences have been rewritten:

The Northeastern region is historically marked by strong social inequality, with a population of approximately 53 million people. It is the region with the largest number of units of the Brazilian federation, nine in total, and with an area equivalent to around 18% of Brazilian territory

13. I find some drawbacks in the sentence on line 110. The beginning is redundant (“were collected” is mentioned twice). I don't understand what “(3)” represents. I suggest clarifying what the "other independent variables" are.

Response: Thank you for your comments. The sentences have been rewritten:

Secondary data were collected from SUS Data Department (DATASUS), among them the study outcomes: confirmed cases and deaths by COVID-193 and some independent variables. Other data were collected in the database of the Brazilian Institute of Geography and Statistics (IBGE), which was compiled by the Brazilian agency of the United Nations Development Programme (UNDP)16.

As for the other independent variables, they are described in the paragraph of line 113:

The time series data refer to the period from February 26 to May 16, 2020. The study primary outcome was COVID-19 cases diagnosed and secondary outcome was deaths by COVID-19. The independent variables were time, analyzed through epidemiological weeks; PHC coverage, Family Health Strategy (FHS) coverage through the proportion of ESF services in PHC in March 2020; evaluation score of the PHC health services quality by the Quality Monitoring and Evaluation Program (PMAQ); number of beds for hospitalization in April 2020; the Human Development Index (HDI), the Gini index, Gross Domestic Product (GDP) per capita, distance to the epicenter of the pandemic in Brazil, Demographic Density, percentage of employed persons and the percentage of social isolation in the nine Federative Units.

14. Line 119. What is ESF?

Response: Thank you for your comments. The sentence was rewritten:

The independent variables were time, analyzed through epidemiological weeks; PHC coverage, Family Health Strategy (FHS) coverage through the proportion of ESF (Family Health Strategy) services in PHC in March 2020; evaluation score of the PHC health services quality by the Quality Monitoring and Evaluation Program (PMAQ); number of beds for hospitalization in April 2020; the Human Development Index (HDI), the Gini index, Gross Domestic Product (GDP) per capita, distance to the epicenter of the pandemic in Brazil, Demographic Density, percentage of employed persons and the percentage of social isolation in the nine Federative Units.

15. Line 180. Is the average taken between cities? Or maybe days?

Response: Thank you for your comments. The sentence was rewritten:

In the nine cities analyzed, we identified an accumulated of 43,969 cases and 2,588 deaths due to COVID-19 up to the twentieth epidemiological week. The average number of diagnosed in the cities evaluated cases was 4885.44 (± 1574.85), 287.55 (± 125.80) of deaths and an overall lethality of 4.4%.

All changes made to the text are highlighted in the manuscript.

Thank you for your comment. The manuscript has been revised accordingly. 

Sincerely,

Sanderson José Costa de Assis. Federal University of Rio Grande do Norte. Corresponding author. Natal, Rio Grande do Norte, Brazil.

Mobile: +5584996219425

e-mail: sanderson_assis@hotmail.com

---

## [Decision Letter · Decision Letter 1]

8 Apr 2021

PRIMARY HEALTH CARE AND SOCIAL ISOLATION AGAINST COVID-19 IN NORTHEASTERN BRAZIL: ECOLOGICAL TIME-SERIES STUDY

PONE-D-20-19401R1

Dear Dr. Assis,

We’re pleased to inform you that your manuscript has been judged scientifically suitable for publication and will be formally accepted for publication once it meets all outstanding technical requirements.

Kind regards,

Lidia Adriana Braunstein, PhD

Academic Editor

PLOS ONE

Additional Editor Comments (optional):

Reviewers' comments:

Reviewer's Responses to Questions

**Comments to the Author**

1. If the authors have adequately addressed your comments raised in a previous round of review and you feel that this manuscript is now acceptable for publication, you may indicate that here to bypass the “Comments to the Author” section, enter your conflict of interest statement in the “Confidential to Editor” section, and submit your "Accept" recommendation.

Reviewer #1: All comments have been addressed

Reviewer #2: All comments have been addressed

Reviewer #3: All comments have been addressed

Reviewer #4: All comments have been addressed

2. Is the manuscript technically sound, and do the data support the conclusions?

Reviewer #1: Yes

Reviewer #2: Yes

Reviewer #3: Yes

Reviewer #4: Yes

3. Has the statistical analysis been performed appropriately and rigorously? 

Reviewer #1: Yes

Reviewer #2: Yes

Reviewer #3: Yes

Reviewer #4: Yes

4. Have the authors made all data underlying the findings in their manuscript fully available?

Reviewer #1: Yes

Reviewer #2: Yes

Reviewer #3: Yes

Reviewer #4: Yes

5. Is the manuscript presented in an intelligible fashion and written in standard English?

Reviewer #1: Yes

Reviewer #2: Yes

Reviewer #3: Yes

Reviewer #4: Yes

6. Review Comments to the Author

Reviewer #1: The manuscript complies with what was requested. Congratulations on your review. It was contemplated in relation to the authors' review.

Reviewer #2: (No Response)

Reviewer #3: The authors made the changes suggested for the manuscript, therefore, I have no further considerations to make.

Reviewer #4: The authors have answered all my questions satisfactorily and I believe that the manuscript was improved enough to be published, for which my recommendation to the editor is to accept the submitted work.

7. PLOS authors have the option to publish the peer review history of their article (what does this mean?). If published, this will include your full peer review and any attached files.

Reviewer #1: **Yes: **Marcelo Cardoso de Souza

Reviewer #2: No

Reviewer #3: No

Reviewer #4: No

---

## [Editor Report · Acceptance letter]

16 Apr 2021

PONE-D-20-19401R1 

Primary health care and social isolation against COVID-19 in Northeastern Brazil: Ecological time-series study 

Dear Dr. Costa de Assis:

I'm pleased to inform you that your manuscript has been deemed suitable for publication in PLOS ONE. Congratulations! Your manuscript is now with our production department. 

Kind regards, 

on behalf of

Dr. Lidia Adriana Braunstein 

Academic Editor

PLOS ONE